# Weakly-supervised & Uncertainty-aware 3D Gaze Estimation with Geometry-guided Constraints

## Abstract

3D eye gaze estimation from monocular images remains to be a challenging task due to the model sensitivity to illumination, occlusion and head pose changes. As the growing interests and demand in in-the-wild 3D gaze estimation under unconstrained environments, the generalization ability has been considered as a crucial performance metric of 3D gaze estimation models. In this work, we present UGaze-Geo, an uncertainty-aware weakly-supervised framework for 3D gaze estimation. We leverage the general knowledge of human eyeball anatomy and develop multiple geometric constraints. The proposed geometrical constraints contains two types, where the first type is formulated by constructing the mapping function from anatomical 3D eyeball parameters to eye appearance features (eyelid & iris landmarks). The second type of constraints is based on the relationship among head rotation, eyeball rotation and gaze, where we learn a variable that describes "relative eyeball rotation" conditioned on current head pose. Both type of constraints are free of gaze labels and are general to any subjects and environmental conditions. We formulate these constraints as loss functions in a probabilistic framework. We evaluate the UGaze-Geo framework on within-domain and four cross-domain gaze estimation tasks to validate the effectiveness of each constraint and the advantage of performing probabilistic gaze estimation. Experimental results indicate that our model achieves SOTA performances on different dataset.

## 1 Introduction

Eye gaze is an important cue for human behaviour and attention analysis. With the growing popularity in interactive applications such as AR/VR, 3D avatar animation, human-computer interaction and driver behaviour monitoring, automatic gaze estimation methods are proposed to regress 3D gaze directions from eye images. With the development deep learning algorithms, CNN models have been fully utilized to directly regress gaze from images and can generate accurate results given well-annotated gaze datasets (Kellnhofer et al., 2019; Zhang et al., 2020; Funes Mora et al., 2014; Fischer et al., 2018). In spite of the good within-dataset performances achieved by recently-proposed learning based gaze estimation methods such as (Cheng et al., 2020; Chen & Shi, 2018; Fischer et al., 2018; Zhang et al., 2017a), these models may suffer from sensitivity to different head poses, illumination conditions and environmental changes when switch to different datasets. Improving the robustness and generalization ability of gaze estimators becomes an emerging topic with great application significance. Gaze360 (Kellnhofer et al., 2019) provides a solution for robust 3D gaze tracking by training the gaze model with the large-scale unconstrained images collected under various environments for various subjects. To further improve the generalization ability of gaze estimation models, researchers have been exploring target domain adaptation (Bao et al., 2022; Cheng et al., 2022; Cai et al., 2020; Liu et al., 2021; Wang et al., 2022; Cui et al., 2020), for updating the pre-trained model with label or unlabeled target samples. On the other hand, 3D model-based methods (Wood et al., 2016; 2018; Wang & Ji, 2017; Ploumpis et al., 2020) have been studied as an alternative to appearance-based methods. Eye-Model-based methods usually solve gaze direction by calculating anatomical eye parameters from eye image features (such as eye landmarks, iris boundary and limbus boundary) and then solving the orientation of eyeball relative to camera. Earlier model-based methods (Hutchinson et al., 1989; Fuhl et al., 2016; Fuhl, 2021; Fuhl et al., 2017) pro-

posed to analyze eye parameters from infrared eye image and can recover very accurate anatomical eye structure. Later, this process is simplified with the advent of 3D deformable eye models (Wood et al., 2016; Ploumpis et al., 2020) and the complex eye parameter calculation can be replaced by fitting the 3D eye bases. The 3D eye model provides a general shape prior that can applied to any web-camera images. However, the process of model-based fitting can be time-consuming and need a pre-calibration procedure for each subject.

To build a robust 3D gaze estimator that can be applied in unconstrained environments, we propose to incorporate general eye structural information into the learning process and formulate them as geometrical constraints. Firstly, we introduce geometric structure of a 3D eyeball, with its shape parameterized by two learnable anatomical variable: eyeball radius $r_e$ and iris radius $r_i$.(a). Along with gaze prediction, our model constructs the correspondences between 3D eyeball vertices and iris & eyelid landmarks by camera projection. Secondly, according to head-eye anatomy, the eyeball center $C_e$ is a fixed location inside the head for each subject and the eyeball can rotate independently from the head around the eyeball center. Therefore, a gaze vector $g$ is a combination of head pose $R_h$ and eyeball movement $R_e$. Directly estimating $g$ from images can be inaccurate due to appearance ambiguity, especially in blurry, occluded or less-illuminated images. The advantage of decomposing gaze into head pose and eyeball movement is that the head pose is an easier term to solve while the eyeball movement $R_e$ (under arbitrary head poses) tends to locate in a certain interval, which is a general anatomical prior. Biomedical study from (Moon et al., 2020) reveals the statistics that the horizontal and vertical movement of eyeball ranges in $[-34.5°, 33.3°]$ and $[-23.4°, 20.1°]$ respectively. Our original contributions of this work can be concluded as follows:

- we propose an uncertainty-aware 3D gaze estimation method UGaze-Geo, where the head pose and eyeball movement are disentangled by learning a probabilistic eyeball rotation function conditioned on head pose.

- we incorporate the geometry knowledge of 3D eyeball in the model, and further propose three anatomy-based geometric constraints, which can be used for weakly-supervised training or semi-supervised training (when gaze labels are available). Specifically. we propose a novel rotation consistency constraint based on the relationship of head rotation and gaze direction.

- experimental results show the effectiveness of the geometrical constraints, in terms of SOTA within-dataset performance and better generalization ability in cross-dataset evaluation than SOTA learning-based models. Our model can also quantify the predicted gaze uncertainty.

## 2 RELATED WORKS

### 2.1 HEAD POSE & EYE POSE DISENTANGLEMENT

The strategy of disentangling head pose and eye pose is often used in generative models, which can be trained on head and gaze labels. (Park et al., 2019) propose to train an encoder-decoder to disentangle gaze direction, head pose and other appearance factors. An embedding consistency loss is applied on frontalized latent gaze features. (Xia et al., 2020) propose a continuous gaze interpolation framework through decoupling related gaze attributes (head pose, vertical and horizontal gaze directions). In addition, multi-view gaze representation learning conducted by (Gideon et al., 2022) provides a solution for disentangling head pose and relative gaze feature given camera information.

### 2.2 REGRESSION OF 3D GAZE WITH CONSTRAINTS

Fully-supervised learning based models rely on the quantity and quality of gaze labels to produce reliable performance. However, generating accurate gaze labels is time consuming and labor expensive. Researchers have proposed different constraints as weak supervision or regularization. One type of constraints is to utilize the geometric knowledge of eyeball models in the training process. (Ploumpis et al., 2020) constructed an anatomical 3D deformable eye model consists of PCA bases of eye socket, pupil size and pupil texture. They train a CNN to predict anatomical eyeball parameters. (Park et al., 2018) pre-train an hourglass-based network to predict the landmarks of iris, pupil center, eyeball center & radius on synthetic dataset, followed by a regressor to predict gaze from

the landmarks. Another commonly used constraint is imposed by the relationship among head pose, eyeball movement and gaze direction. (Zhu & Deng, 2017) trained separate head pose and eye pose estimation models and utilized a gaze transform layer to convert head pose and eyeball movement to gaze direction. In the work proposed by (Kothari et al., 2021), 3D head location and orientation are used to impose strong gaze-related geometric constraints from two people "Looking At Each Other", which serves as a weak supervision term on the spatial relationship of gaze vectors so their model can be trained without gaze labels. Our model combines anatomical knowledge of 3D eyeball structure with head-eye-gaze relationships, so they can be incorporated into the training as regularization terms. Unlike (Zhu & Deng, 2017), whose geometric constraints can only be imposed on constrained images (i.e., in-lab images with regular head poses) with given head pose and eyeball movement labels, our model figures out the relationship among $g, R_h, R_e$ automatically without requiring any additional eyeball movement labels and can be applied on unconstrained in-the-wild images.

### 2.3 Gaze synthesizing & augmentation

Eye image synthesis or augmentation are common ways to enrich training data, especially when real gaze labels are hard to obtain. Synthesizing eye images from generative models (Wang et al., 2018; Kaur & Manduchi, 2021; Yu et al., 2019) helps improve gaze estimation accuracy with more diverse and gaze-controllable data. Meanwhile, it is straight-forward to conduct data augmentation by applying image transformations, such as the 2D rotation in the work of Bao et al. (Bao et al., 2022) and 3D warping implemented by Qin et al. (Qin et al., 2022). Like (Qin et al., 2022), we generate 3D head pose augmentations on a gaze dataset, but we step further, by applying the anatomic knowledge of eyeball and head-eye-gaze relationships on the the augmented data.

## 3 Methods

We describe the workflow of our method as below. Our model first estimates head pose $R_h$ and then predict a probabilistic eyeball movement distribution conditioned on current head pose feature, i.e., $p(R_e|R_h, I)$. At last, gaze direction is represented by a combination of head pose $R_h$ and eyeball movement $R_e$. This section consists of the following part. In Section 3.1 we introduce the geometry-guided constraints, including the geometric eyeball model, transformation of coordinate systems and the relationship among head pose, eyeball movement and gaze. In Section 3.2, we describe process of training a deterministic model with the constraints , denoted as Gaze-Geo. In Section 3.3, we describe the uncertainty-aware Gaze-Geo framework, abbreviated as UGaze-Geo, including re-writing of the probabilistic loss terms and uncertainty analysis. Finally, in Section 3.4, we describe the full training loss of UGaze-Geo.

### 3.1 Preliminary: Geometry-guided Constraints

In this section, we will describe the anatomy-based geometric constraints applied in training. The first type of constraint is based on eye-anatomy. According to (Wang & Ji, 2017; Wang et al., 2018), 3D eyeball geometry can be estimated by a two-sphere model, with a larger sphere representing the eyeball and a smaller one representing the cornea sphere. The intersection boundary of the two spheres forms the boundary of iris. We use two parameters {eyeball radius: $r_e$, iris radius: $r_i$} to describe the shape of 3D eyeball. Then the eyeball center $C_e$ and iris center $C_i$ are used to describe the location of eyeball and iris. We impose two constraints based on eye-anatomy: (1) the 3D eyelid vertices, which can be easily detected by existing 3D facial landmarks detectors (Bulat & Tzimiropoulos, 2017), should be nearly located on the surface of 3D eyeball; (2) the 3D iris boundary & iris center should be well aligned to the iris in image after proper camera projection.

The second type of constraints are based on head-eye anatomy. One general assumption according to anatomical knowledge is that the eyeball center location $C_e$ in head is fixed, and eyeball can only rotate around $C_e$ within certain range of degrees along pitch axis (horizontal) and yaw axis (vertical). Based on this assumption, we decompose the gaze direction into the global head rotation $R_h$ and the secondary eyeball movement $R_e$, where $R_e$ is relative eyeball rotation conditioned on $R_h$. $R_h$ and $R_e$ are $3 \times 3$ rotation matrices converted from rotation angles. Since $R_e$ is converted

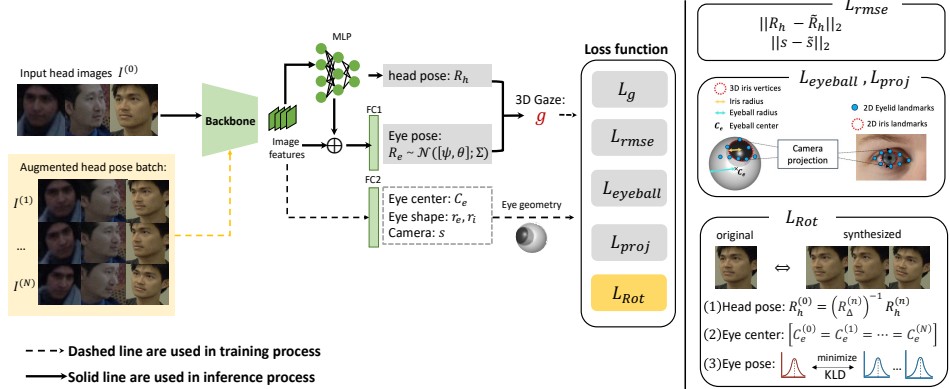

Figure 1: Overview of our model structure and training process. *Green part*: the model structure for performing gaze estimation, including a ResNet18 CNN, a MLP for regressing 3D head pose and two fully connected layers for regressing eye pose and eye shape parameters respectively. *Yellow part*: the head-pose augmented images we generated for the training process and the corresponding geometric rotation consistency loss $L_{Rot}$. Steps with dashed arrows can be skipped during inference.

from two rotation angles, pitch $\psi$ and yaw $\theta$, it can be written as below:

$$R_e = R_y(\theta)R_x(\psi) = \begin{bmatrix} \cos\theta & \sin\psi\sin\theta & \cos\psi\sin\theta \\ 0 & \cos\psi & -\sin\psi \\ -\sin\theta & \sin\psi\cos\theta & \cos\psi\cos\theta \end{bmatrix} \quad (1)$$

Then the gaze direction relative to camera can be written by combining head pose and eyeball movement, denoted as $\hat{g} = R_e R_h [0, 0, -1]^T$. Changing one element in $R_h$ and $R_e$ and keep another fixed will result in various appearances and gaze. In this paper we synthesize mini batches of images with different head poses and the same eyeball movement for imposing the rotation consistency constraint, which is no matter how the head rotates, the eyeball center location $C_e$ in head is always unchanged.

### 3.2 METHOD DESCRIPTION: GAZE-GEO

**Method Description for Gaze-Geo** We first introduce the deterministic framework Gaze-Geo without considering the uncertainty. The overview of designed network structure and training process are illustrated in Fig. 1. Our network takes face images as input, regressing for three sets of parameters: $\{R_h, s\}$, $\{C_e, r_e, r_i\}$ and $\{\psi, \theta\}$.

We first describe two basic loss terms for training, then we will discuss loss functions derived from anatomy-based geometric constraints as regularization terms.

**MSE error on head parameters** The head pose parameter $R_h$ and camera scaling factor $s$ are the first set of predicted parameters. We apply a fully-supervised RMSE loss defined as:

$$L_{rmse} = \|f(R_h) - f(\tilde{R}_h)\|_2 + \|s - \tilde{s}\|_2 \quad (2)$$

where $f(\cdot)$ function converts rotation matrix to Euler angles, $\tilde{R}_h$ and $\tilde{s}$ are the labels for head pose and camera factor that we prepared in advance from data.

**Gaze loss.** As mentioned in Section. 3.1, given the predicted eye pose $[\theta, \phi]$, we can calculate the eyeball rotation matrix $R_e$ using Eq. 1 and compute gaze direction $g = R_e R_h [0, 0, -1]^T$. If any gaze labels are available during training, we can compute the gaze angular loss as

$$L_g = \|g - g^{gt}\|_1 \quad (3)$$

Then we describe three geometric constraints applied during training.

***Constraint 1: Iris re-projection correspondence*** To fully utilize the geometrical model and lessen the reliance on gaze labels, we construct a iris projection loss function for training our model. Defining the z-axis as vertical to the iris plane in eyeball coordinate system, the iris points $z_{iris}^{3d}$ in ECS

can be expressed as:

$$\begin{cases} (z_{iris}^{3d}[1])^2 + (z_{iris}^{3d}[2])^2 = r_i{}^2 \\ \qquad z_{iris}^{3d}[3] = \sqrt{r_e{}^2 - r_i{}^2} \end{cases} \tag{4}$$

We can uniformly sample $K$ points on the iris circle. With predicted head and eyeball parameters, the projected iris points can be calculated by:

$$z_{iris}^{2d} = sP(R_e R_h z_{iris}^{3d} + R_h C_e) + sT_h \tag{5}$$

We have a pre-trained iris detector for automatically generating $K$ iris landmarks and pupil center landmark from eye images, which are used as annotations. The overall $K + 1$ landmarks are denoted by $z_{iris}^{2d} = \{(x_i, y_i)\}_{i=1,\cdots,K+1}$. Considering that the resolution of eye images vary among different datasets (for example, eye images from Gaze360 are quite blurred and hard to detect iris), the iris detector outputs a 2D Gaussian distribution for each landmark location, i.e., $p(\tilde{x}_i, \tilde{y}_i) = p(\tilde{z}_i | I) = \mathcal{N}(\tilde{z}_i; u_{\tilde{z}_i}, \Sigma_{\tilde{z}_i})$. We performed a study of how iris annotation accuracy affect the accuracy of UGaze-Geo, which are discussed in *Section B.3, Appendix*.

To minimize the landmark projection error, we construct the negative log-likelihood loss of projected $z_{iris}^{2d}$ as

$$L_{proj} = -\frac{1}{K+1} \sum_i^{K+1} \log(\mathcal{N}(z_{iris}^{2d}[i]; u_{\tilde{z}_i}, \Sigma_{\tilde{z}_i})) \tag{6}$$

***Constraint 2: 3D eyelid and eyeball radius***  In order to avoid unreasonable eyeball size, we further apply a geometric constraint that all the 3D eyelid landmarks should be located near the surface of eyeball. The eyeball radius regularization loss is defined as

$$L_{eyeball} = \|d(z_{eyelid}^{3d}, C_e) - (1 + \sigma)r_e\|_2 \tag{7}$$

where $d(*, *)$ is the Euclidean distance function and $\sigma$ is a pre-defined hyper-parameter with small positive value.

***Constraint 3: Model Rotation Consistency***  We manually generated $N$ head-pose-augmented images for each training image, by multiplying a random rotation matrix $R_\Delta^{(n)}, n = 1, \cdots, N$ on the original head pose and synthesizing a new image $I^{(n)}$ by 3D warping. Examples of augmented batch are shown in Fig. 5 in Appendix. Detailed steps of generating augmented images are discussed in *Section B.1, Appendix*. Then we propose rotation consistency constraints between original image and the batch of augmented data. Our model takes the batch of the original image $I^{(0)}$ with $N$ augmented images $\{I^{(n)}\}_{n=1}^N$ as input. The constraints are described below, formulated as loss terms. (1) head pose $R_h^{(n)}$ should be consistent with $R_h^{(0)}$ after multiplying the inverse delta rotation matrix $(R_\Delta^{(n)})^{-1}$

$$L_{Rot-h} = \frac{1}{N} \sum_{n=1}^N (\|f(R_h^{(0)}) - f((R_\Delta^{(n)})^{-1} R_h^{(n)})\|_2) \tag{}$$

(2) eyeball center position in HCS should be a constant value no matter how the head rotates:

$$L_{Rot-c} = \frac{1}{N} \sum_{n=1}^N \|C_e^{(n)} - C_e^{(0)}\|_2 \tag{}$$

(3) relative eyeball rotation angle $\{\psi, \theta\}$ should be consistent across the batch since no additional eyeball movement is involved.

$$L_{Rot-e} = \frac{1}{N} \sum_{n=1}^N \|[\psi^{(0)}, \theta^{(0)}] - [\psi^{(n)}, \theta^{(n)}]\|_2 \tag{}$$

The model rotation consistency loss is defined as follows:

$$L_{Rot} = L_{Rot-h} + L_{Rot-c} + L_{Rot-e} \tag{8}$$

The total loss function for training the Gaze-Geo model is:

$$L = \lambda_1 L_{rmse} + \lambda_2 L_{proj} + \lambda_3 L_{eyeball} + \lambda_4 L_g \tag{9}$$

### 3.3 UGAZE-GEO: UNCERTAINTY-AWARE VERSION OF GAZE-GEO

Due to image blur or self-occlusion caused by head pose movement, it can be hard to observe a clear iris boundary or eye appearance change from an input eye image. We train a CNN to predict a Gaussian distribution $\mathcal{N}(\boldsymbol{u}_e, \boldsymbol{\Sigma}_e)$ rather than deterministic output $\{\psi, \theta\}$. Then we have

$$p(\psi, \theta | I, R_h) = \mathcal{N}([\psi, \theta]^T; \boldsymbol{u}_e, \boldsymbol{\Sigma}_e)$$

$$\boldsymbol{u}_e, \boldsymbol{L}_e = \mathcal{F}_3\left(\left[\mathcal{G}(I^{(n)}), R_h^{(n)}\right]\right), \boldsymbol{\Sigma}_e = \boldsymbol{L}_e \boldsymbol{L}_e^T \tag{10}$$

where $\mathcal{F}_3, \mathcal{G}$ represents network weights of CNN + FC3 as shown in Fig. 1, the output will be a two-dimensional mean vector $\boldsymbol{u}_e = [u_\psi, u_\theta]$ and the Cholesky decomposition matrix $\boldsymbol{L}_e$ such that the covariance matrix $\boldsymbol{\Sigma}_e = \boldsymbol{L}_e \boldsymbol{L}_e^T$. With the predicted distribution for eyeball rotation angles, we can re-write two loss functions, Eq. 6 and Eq. 8, which are related with eyeball rotations.

**Re-write1: iris re-projection loss on samples**   Since we do not have ground-truth eyeball rotation angles, to make sure that the estimated rotation distribution matches with the visual evidence in the eye images, we re-write the iris re-projection loss in Eq. 6 by sampling rotation angles from the predicted distribution in Eq. 10. We can generate $S$ samples $\{[\hat{\psi}_j, \hat{\theta}_j] \sim \mathcal{N}([\psi, \theta]^T; \boldsymbol{u}_e, \boldsymbol{\Sigma}_e)\}_{j=1}^S$ using re-parameterization trick introduced by (Kingma & Welling, 2013). Constructing sample-based re-projection loss is formulated as below.

$$L_{proj} = -\frac{1}{S(K+1)} \sum_{j=1}^{S} \sum_{i=1}^{K+1} \log(\mathcal{N}(\hat{\boldsymbol{z}}_{iris,j}^{2d}[i]; \boldsymbol{u}_{\tilde{z}_i}, \boldsymbol{\Sigma}_{\tilde{z}_i}))$$

$$\text{where } \hat{\boldsymbol{z}}_{iris,j}^{2d} = sP(\hat{R}_{e(j)} R_h \boldsymbol{z}_{iris}^{3d} + R_h C_e) + sT_h \tag{11}$$

$$\hat{R}_{e(j)} = R_y(\hat{\theta}_j) R_x(\hat{\psi}_j)$$

**Re-write2: eyeball rotation consistency loss.**   On augmented images, we apply "$L_{Rot-e}$" in Eq. 8 to constrain that the eyeball rotation be consistent with original image. With predicted distribution $P^{(n)} \triangleq p(\psi, \theta | I^{(n)}, R_h^{(n)}) = \mathcal{N}([\psi, \theta]^T; \boldsymbol{u}_e^{(n)}, \boldsymbol{\Sigma}_e^{(n)})$, we can re-write RMSE loss $L_{Rot-e}$ as a KL-Divergence loss:

$$L_{Rot-e} = \frac{1}{N} \sum_{n=1}^{N} (KL(P^{(n)} \mid P^{(0)}))$$

$$= \frac{1}{2}[\log \frac{|\boldsymbol{\Sigma}_e^{(0)}|}{|\boldsymbol{\Sigma}_e^{(n)}|} + \text{tr}((\boldsymbol{\Sigma}_e^{(0)})^{-1} \boldsymbol{\Sigma}_e^{(n)}) \tag{12}$$

$$+ (\boldsymbol{u}_e^{(n)} - \boldsymbol{u}_e^{(0)})^T (\boldsymbol{\Sigma}_e^{(0)})^{-1} (\boldsymbol{u}_e^{(n)} - \boldsymbol{u}_e^{(0)})] - 1$$

**Uncertainty of Gaze**   As described in Eq. 10 and **re-write1**, we can compute the final output as the sample average of rotation angles: $[\bar{\psi}, \bar{\theta}] = \frac{1}{S} \sum_{j=1}^{S} [\psi_j, \theta_j]$. Then the eyeball rotation can be calculated as $\bar{R}_e = R_y(\bar{\theta}) R_x(\bar{\psi})$, then 3D gaze direction is expressed as

$$\bar{g} = \bar{R}_e R_h [0, 0, -1]^T \tag{13}$$

We can also compute the variance of gaze as:

$$\text{Var}_g = \sum_{j=1}^{S} sph(\hat{R}_{e(j)} R_h [0, 0, -1]^T) \tag{14}$$

where $sph(\cdot)$ is the function that converts a Cartesian coordinate to spherical coordinates (azimuth, elevation). In Eq. 14, $\text{Var}_g$ contains two variances $[\sigma_{azi}, \sigma_{ele}]$ representing azimuthal and elevation uncertainty respectively.

### 3.4 UGAZE-GEO: FINAL TRAINING LOSSES

After introducing the model rotation consistency loss and uncertainty-based loss into the baseline model, we have the final loss function for the proposed UGaze-Geo method:

$$L = \lambda_1 L_{rmse} + \lambda_2 L_{proj} + \lambda_3 L_{eyeball} + \lambda_4 L_{Rot} + \lambda_5 L_g \tag{15}$$

Table 1: We train Gaze-Geo with different percentages of labels in Gaze360 and evaluate them on the same testing data. Row1 and Row2 are reference methods training on full data of Gaze360.

| Models | Within-data | | Cross-data | |
|---|---|---|---|---|
| | Test (frontal) | Test (full) | $\mathcal{D}_G \rightarrow \mathcal{D}_M$ | $\mathcal{D}_G \rightarrow \mathcal{D}_D$ |
| (Kellnhofer et al., 2019) 100% training data | 11.1 | 13.5 | 11.36 | 11.86 |
| (Bao et al., 2022) 100% training data | - | - | 7.60 | 7.10 |
| ours:100% gaze label | 10.03 | 10.87 | 7.57 | 6.98 |
| ours:50% gaze label | 10.89 | 11.32 | 7.83 | 7.08 |
| ours:25% gaze label | 11.98 | 13.15 | 8.56 | 8.22 |
| ours:0% gaze label | 16.62 | 19.74 | 17.86 | 27.20 |

with $L_{rmse}$ defined in Eq. 2 , $L_{nll}$ defined in Eq. 11, $L_{eyeball}$ defined in Eq. 7, $L_{Rot}$ defined in Eq. 8 and Eq. 12, and $L_g = \|\bar{g} - g^{gt}\|_1$ when gaze label is available. We summarize the training process in Algorithm 1 in Appendix.

## 4 EXPERIMENTS

**Datasets.** We investigate model performance on four benchmark datasets: Gaze360(Kellnhofer et al., 2019) ($\mathcal{D}_\mathcal{G}$), ETH-XGaze (Zhang et al., 2020)($\mathcal{D}_\mathcal{E}$), MPIIFaceGaze (Zhang et al., 2017b) ($\mathcal{D}_\mathcal{M}$)and EyeDiap (Funes Mora et al., 2014)($\mathcal{D}_\mathcal{D}$). 1) Gaze360 contains in-the-wild human images captured by a 360° camera with a wide range of horizontal gaze direction. Following the setting in (Bao et al., 2022), we select 84900 images for data augmentation and model training. 2) ETH-XGaze contains high-resolution face images of 80 participants collected from a multi-camera system in a laboratory environment. 3) MPIIFaceGaze provides human face images captured by a laptop camera when a participant is looking at a target on the screen, containing data for 15 participants in total. 4) EyeDiap is a video-based dataset recording a participant head and eye movement when tracking a static or a moving target.

**Training data augmentation.** We introduce the data augmentation process at the beginning of Section 3.2. In experiments we generate N = 5 augmented head rotation matrices. More details about generating augmented images are described in the Appendix.

**Training Details.** For gaze prediction, we generate 10 samples (i.e., $S = 10$ in Eq. 11) from predicted distribution. To study the effectiveness of the geometry-guided constraints, our model is trained with 0%, 25%, 50% and 100% of gaze labels respectively.

### 4.1 WEAKLY- AND SEMI-SUPERVISED GAZE LEARNING

The proposed constraints can be directly used for weakly-supervised learning on any dataset or combined with gaze labels to perform semi-supervised learning. We train the model with less data from Gaze360. As the full training labels of Gaze360 is ∼85k, we use 50% (∼42k), 25% (∼21k) and 0% of them and train the model respectively. We present the corresponding performances in Table. 1. In the last three rows we show the within- and cross-data performance of our Gaze-Geo model trained on full, 50% and 25% of images of Gaze360. When using 50% of images, Gaze-Geo still outperforms (Kellnhofer et al., 2019) on Gaze360 testing set, for both frontal poses or full poses; and our model achieves comparable cross-data performances with (Bao et al., 2022). When reducing the training data size to 25%, the gaze estimation accuracy declined but still in an acceptable range (better than (Kellnhofer et al., 2019)). Based on the within- and cross-data evaluations, we can prove that training with the proposed anatomy-based constraints helps to encode the general anatomical knowledge about eyeball movement (conditioned on head pose) into the model. Therefore, our model is data efficient and can be trained with less labels but can still maintain reasonably good generalization ability.

### 4.2 FULLY-SUPERVISED LEARNING: WITHIN-DATASET EVALUATION

To perform a fair comparison and prove that our proposed geometric constraints improve gaze estimation performance, we first perform within-dataset evaluation on Gaze360 and MPIIFaceGaze, as

Table 2: Within-dataset evaluation

| Methods | Gaze360 (frontal) | Gaze360 (full) | MPIIFaceGaze |
|---|---|---|---|
| RT-Gene (Fischer et al., 2018) | 12.26 | - | 4.3 |
| Dilated-Net (Chen & Shi, 2018) | 13.73 | - | 4.8 |
| CA-Net (Cheng et al., 2020) | 12.26 | - | 4.1 |
| Gaze360 (Kellnhofer et al., 2019) | 11.4 | 13.5 | - |
| LAEO (Kothari et al., 2021) | 10.1 | 13.2 | - |
| L2CS (Abdelrahman et al., 2022) | 10.41 | - | 3.92 |
| **Gaze-Geo(ours) + 100% labels** | **10.03** | ***10.87** | **3.71** |
| **Gaze-Geo(ours) + 50% labels** | **10.89** | **11.32** | **3.78** |
| **UGaze-Geo(ours) + 100% labels** | ***9.88** | **11.06** | ***3.66** |

Table 3: Cross-dataset evaluation and comparision with SOTA learning-based methods

| Methods | $\mathcal{D}_E \to \mathcal{D}_M$ | $\mathcal{D}_E \to \mathcal{D}_D$ | $\mathcal{D}_G \to \mathcal{D}_M$ | $\mathcal{D}_G \to \mathcal{D}_D$ |
|---|---|---|---|---|
| FullFace (Zhang et al., 2017a) | 12.35 | 30.15 | 11.13 | 14.42 |
| RT-Gene (Fischer et al., 2018) | - | - | 21.81 | 38.60 |
| Dilated-Net (Chen & Shi, 2018) | - | - | 18.45 | 23.88 |
| Gaze360 (Kellnhofer et al., 2019) | 7.23 | 8.02 | 11.36 | 11.86 |
| CA-Net (Cheng et al., 2020) | - | - | 27.13 | 31.41 |
| PureGaze (Cheng et al., 2022) | 7.08 | 7.48 | 9.28 | 9.32 |
| Res-Net18+RAT (Bao et al., 2022) | 7.92 | 7.44 | 7.60 | 7.10 |
| **Gaze-Geo (ours)** | **7.13** | **9.80** | **7.57** | **6.98** |
| **UGaze-Geo (ours)** | ***6.92** | **9.84** | ***7.23** | ***6.87** |

shown in Table 2. We compare our method, the Gaze-Geo and UGaze-Geo, with SOTA learning-based methods, including RT-Gene (Fischer et al., 2018), Dilated-Net (Chen & Shi, 2018), CA-Net (Cheng et al., 2020), Gaze360 (Kellnhofer et al., 2019), LAEO (Kothari et al., 2021) and L2CS (Abdelrahman et al., 2022). On Gaze360 we use the official train-val-test set division and present the evaluation results on different ranges of gaze directions, including frontal faces (column 2 in Table 2) and all faces (column 3 in Table 2). For MPIIFaceGaze, We performed leave-one-out cross validation protocol as (Abdelrahman et al., 2022; Murthy & Biswas, 2021). Our method outperforms other methods on Gaze360, especially on the full range of evaluation set Gaze360(full), Gaze-Geo reduces the gaze angular error by 17.7% comparing with LAEO ((Kothari et al., 2021)). Our model also achieves SOTA performances on MPIIFaceGaze, with 5.4% and 6.6% of reduced gaze error by Gaze-Geo and UGaze-Geo. The row of **Gaze-Geo + 50% labels** shows the within-dataset results of a semi-supervised model trained with 50% gaze labels, and we can still achieve better performances than several SOTA models.

### 4.3 FULLY-SUPERVISED LEARNING: CROSS-DATASET EVALUATION

We also conduct four cross-dataset experiments to elaborate that, by imposing the geometric constraints during training, our gaze estimator is robust and have better generalization ability under large data difference. Following the cross-data settings adopted in existing works PureGaze (Cheng et al., 2022) and RAT (Bao et al., 2022), we train our model respectively on Gaze360 and ETH-XGaze and then evaluate on MPIIFaceGaze and EyeDiap. In Table 3 we compare the performance of our model with SOTA gaze estimation methods, including RT-Gene, Dilated-Net, CA-Net, FullFace, Gaze360, PureGaze and RAT.

As shown in Table 3, our model can achieve three SOTA cross-dataset performance(bold number with a * symbol), with 2.26%, 4.87% and 3.24% improvements on the task of $\mathcal{D}_E \to \mathcal{D}_M$, $\mathcal{D}_G \to \mathcal{D}_M$ and $\mathcal{D}_G \to \mathcal{D}_D$. The lower accuracy on task $\mathcal{D}_E \to \mathcal{D}_D$ occurs possibly because of the image resolution shift, where the 3D eyeball model may be misaligned to lower resolution images in $\mathcal{D}_D$. We have a more detailed analysis in *Section B, Appendix*. Compared to RAT (Bao et al., 2022), which introduced a rotational loss on 2D augmented data, our model can achieve much better performance on three cross-data tasks. It shows that our 3D augmented data + three geometric constraints are effective to improve the model. We show visualizations of predicted gaze and the corresponding geometry (optional in testing) in Fig. 2.

### 4.4 ABLATION STUDY.

We perform an ablation study to validate the effectiveness of our proposed constraints. Our baseline model is the backbone model trained using the $L_g$ and $L_{rmse}$ (the solid line branch in Fig. 1),

Table 4: Ablation study of gaze angular errors when applying different constraints and w/o uncertainty modeling during training. The last two rows are corresponding to Gaze-Geo and UGaze-Geo.

| Models | Within-data | | Cross-data | |
|---|---|---|---|---|
| | $\mathcal{D}_G \rightarrow \mathcal{D}_G$ | $\mathcal{D}_M \rightarrow \mathcal{D}_M$ | $\mathcal{D}_G \rightarrow \mathcal{D}_M$ | $\mathcal{D}_G \rightarrow \mathcal{D}_D$ |
| baseline | 11.97 | 4.40 | 8.90 | 9.66 |
| baseline+Geo-1 | 11.91 | 4.27 | 8.84 | 9.30 |
| baseline+Geo-1,2 | 11.92 | 4.11 | 8.20 | 8.87 |
| baseline+Geo-3 | 11.40 | 3.94 | 7.87 | 7.62 |
| baseline+Geo-1,2,3 | **10.87** | 3.71 | 7.57 | 6.98 |
| baseline+Geo-1,2,3+U | 11.06 | **3.66** | **7.23** | **6.87** |

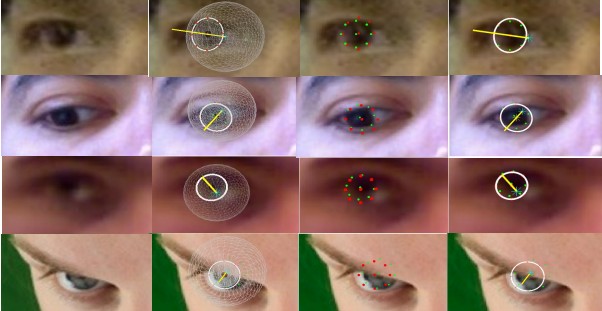

| eye image | projected eyeball (optional in testing) | projected iris (optional in testing) | predicted gaze | 3D gaze from eyeball center |

Figure 2: Visualization of the predicted 3D eyeball, 3D gaze direction and projected iris landmarks on different dataset. The red dots are projected iris vertices and iris center from 3D eyeball while the green dots are iris labels. The yellow line represents the gaze direction.

without applying any constraints or uncertainty modeling. The major difference of our baseline model compared with the static baseline model of Gaze360 (Kellnhofer et al., 2019) is that ours learns a disentangled representation of head pose and eyeball movement from face images. We present the model trained with the i-th constraint as "baseline + Geo-i".

We summarize the results of within & cross dataset evaluation in Table 4. Through comparing the first three rows, we can conclude that the model is well improved with proposed eye-anatomy-based constraints (Geo-1,2). As the two constraints help to optimize the eyeball shape and orientation, we observe smaller gaze error on the last three tasks, as $(4.40° \rightarrow 4.11°)$, $(8.90° \rightarrow 8.20°)$ and $(9.66° \rightarrow 8.87°)$ respectively. Similarly, we verify the importance of the proposed head-eye-anatomy constraints (Geo-3) by applying it singly on baseline, with better gaze accuracy than applying Geo-1,2. Combining all three constraints (baseline+Geo-1,2,3) together helps to achieve a significant improvement on within- and cross-data tasks, with reduced gaze prediction error by 9.2%, 15.7%, 14.9%, 27.8% respectively compared to baseline model. The uncertainty-aware model in the last row shows that with probabilistic modeling, our model can generalize better under cross-data settings, improving the baseline+Geo-1,2,3 model from $(7.57° \rightarrow 7.23°)$, $(6.98° \rightarrow 6.87°)$ on the two cross-data tasks. In addition, we conducted a study of model data efficiency, where we train the model with the geometric constraints but less number of images. We use (full, 50%, 25%) of training images from Gaze360 and evaluate the within- and cross-data performances. We conclude and analyze the results in *Section C, Appendix*. It shows that with the geometric constraints, our model can maintain a relatively good performance with reduced training data.

## 4.5 UNCERTAINTY VALIDATION

We calculate predicted gaze uncertainty by sampling from the estimated rotation distribution, with the gaze vector in 3D space expressed by spherical coordinates $[u_{azi}, u_{ele}]$ and sample standard deviation being $[\sigma_{azi}, \sigma_{ele}]$. We provide detailed discussions and visualizations of gaze uncertainty in *Section E, Appendix*.

## 5 CONCLUSION

In this paper we proposed UGaze-Geo, a novel learning-based framework for 3D gaze estimation, with three geometry-guided geometric constraints applied as wweak-supervision in training. Our model estimates a probabilistic eyeball movement conditioned on head poses and can predict both gaze and gaze uncertainty through aligning 3D eyeball to the eye landmarks. We first introduce two geometric constraints, by mapping a 3D eyeball to eye features including eyelid and iris landmarks. We also propose a model rotation consistency constraint, which is based on augmented data generated by randomly revising the original 3D head pose of an image. With the constraints applied in training, our model is robust to unconstrained images in both within- and cross-dataset evaluations and can achieve SOTA performances.

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

# APPENDIX

## A  TRAINING ALGORITHM

---

**Algorithm 1** Ugaze-Geo Model Training Process

---

1: **Pre-trained model:**
2: Iris-lmk-Net output: $\{\mathcal{N}(\tilde{z}_i; \boldsymbol{u}_{\tilde{z}_i}, \boldsymbol{\Sigma}_{\tilde{z}_i})\}_{i=1}^{K+1}$
3: **Training UGaze-Geo:**
4: Input: Original + aug images:$I^{(0)}, I^{(1)}, \cdots, I^{(N)}$
5: Labels: $\begin{cases} \text{Ground-Truth Gaze: } g^{gt} \\ \text{delta rotation matrix: } R_{\Delta}^{(1)}, \cdots, R_{\Delta}^{(N)} \\ \text{Iris landmarks : } \tilde{\boldsymbol{z}}_i^{(0)}, \cdots, \tilde{\boldsymbol{z}}_i^{(N)} \end{cases}$
6: Model: backbone $\mathcal{G}(\cdot)$; FC-layers $\mathcal{F}_1(\cdot), \mathcal{F}_2(\cdot), \mathcal{F}_3(\cdot)$
7: **for** bs $\leftarrow$ 1 to Max-Batch-Num **do**
8:     **for** n $\leftarrow$ 0 to N **do**
9:         $\{R_h^{(n)}, t_x^{(n)}, t_y^{(n)}, s^{(n)}\} \leftarrow \mathcal{F}_1\left(\left[\mathcal{G}(I^{(n)}), \boldsymbol{z}_{face}^{(n)}\right]\right)$
10:         $\Rightarrow$ compute $L_{rmse}$ by Eq. 2
11:         $\{C_e^{(n)}, r_e^{(n)}, r_i^{(n)}\} \leftarrow \mathcal{F}_2\left(\left[\mathcal{G}(I^{(n)}), R_h^{(n)}\right]\right)$
12:         $\Rightarrow$ compute $L_{eyeball}$ by Eq. 7
13:         $\mathcal{N}([\psi, \theta]^T; \boldsymbol{u}_e^{(n)}, \boldsymbol{\Sigma}_e^{(n)}) \leftarrow \mathcal{F}_3\left(\left[\mathcal{G}(I^{(n)}), R_h^{(n)}\right]\right)$
14:         $\Rightarrow$ compute $L_{Rot}$ by Eq. 8 and Eq. 12
15:         **Sampling:**
16:         $\{[\hat{\psi}_j, \hat{\theta}_j] \sim \mathcal{N}([\psi, \theta]^T; \boldsymbol{u}_e^{(n)}, \boldsymbol{\Sigma}_e^{(n)})\}_{j=1}^S$
17:         $\Rightarrow$ compute $L_{nll}$ by Eq. 11
18:         **Gaze:**
19:         $[\bar{\psi}, \bar{\theta}] = \frac{1}{S}\sum_{j=1}^S [\hat{\psi}_j, \hat{\theta}_j] \Rightarrow \bar{R}_e = R_y(\bar{\theta})R_x(\bar{\psi})$
20:         $\bar{g} = \bar{R}_e R_h[0, 0, -1]^T \Rightarrow L_g = \|\bar{g} - g^{gt}\|_1$
21:     Train $\mathcal{F}_1, \mathcal{F}_2, \mathcal{F}_3, \mathcal{G}$ with Eq. 15.

---

## B  TRAINING DATA PREPARATION

Our model contains a training data preparation process, including image augmentation, label generation for head pose and iris-landmark localization, as illustrated in Fig. 4. We describe each part in detail in the following sections.

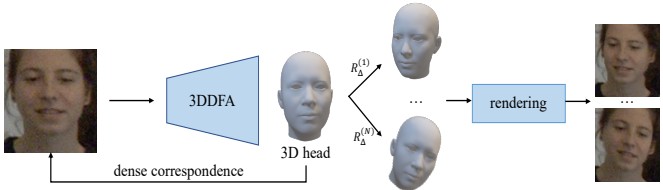

Figure 3: 3D head augmentation pipeline.

### B.1  AUGMENTING EYE IMAGES WITH VARIOUS HEAD POSES

We refer to Guo et al. (Guo et al., 2020) to prepare the augmented gaze data. The process is described in Fig. 3. The augmentation only entails head orientation changes without involving any additional eyeball rotation. Therefore, it's reasonable to introduce a consistency constraint on 3D relative eyeball rotations between an original image and all of its augmented images. On Gaze360 and

MPIIFaceGaze, we randomly generate five delta rotation angles of pitch, yaw and roll, given the head pose of each original image. The data augmentation procedure works well under different illumination, image resolution and head poses. We provide examples of synthesized images in Fig. 5. On ETH-XGaze, since it is a multi-view dataset, we do not further augment additional head poses. We take the images from one camera as the reference image, then extract the head pose differences in pitch and yaw angles from other cameras. The group of images from multiple camera can be considered as head variations with the same relative eye pose so that we can apply the proposed model rotation consistency constraints.

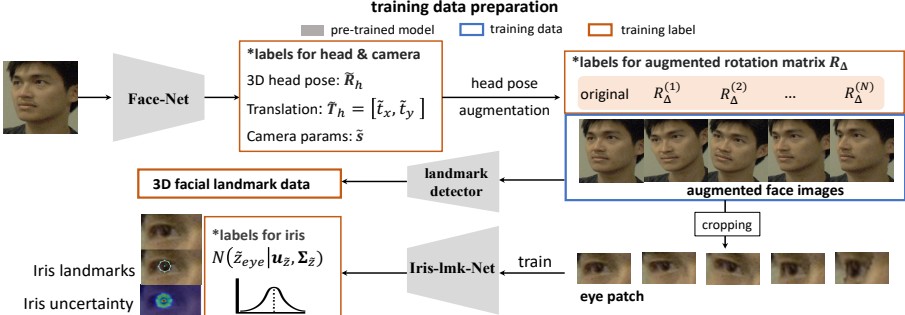

Figure 4: Training data and label preparation through pre-trained models (grey modules in figure). 1) We use a re-trained 3DDFA Guo et al. (2018) as **Face-Net** for generating 3D head pose and camera parameter labels as $\tilde{R}_h, \tilde{t}_x, \tilde{t}_y, \tilde{s}$ for the original training image. 2) Then based on the head pose $\tilde{R}_h$ we further synthesize different head pose views by augmenting the rotation angles in pitch, yaw and roll angles, resulting in the augmented rotation matrix label $R_\Delta$. 3) We generate 3D facial landmark data on the augmented face images by using the package of Bulat & Tzimiropoulos (2017) and crop them to eye region data. 4) We explicitly train a model **Iris-lmk-Net** that predicts a distribution for iris landmarks $\mathcal{N}(\tilde{z}_{eye}|\mu_{\tilde{z}}, \Sigma_{\tilde{z}})$ on eye patch data. The mean value $\mu_{\tilde{z}}$ are used as iris landmark labels and the variance $\Sigma_{\tilde{z}}$ will be used to weight the iris re-projection loss (higher variance, lower weight to the loss), which can mitigate the impact of inaccurate landmarks due to illumination, image resolution or occlusion.

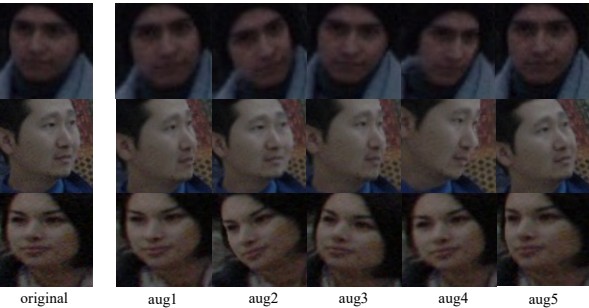

Figure 5: Synthesized 3D head augmentation examples under different illumination, resolution and head poses.

## B.2 FACE-NET FOR HEAD POSE ALIGNMENT

We re-train the model 3DDFA (Bulat & Tzimiropoulos, 2017) following the code provided by (Guo et al., 2018) and perform dense 3D face alignment on the training data. For those training images without head pose labels, we use the output of this model as the labels for 3D head poses $\tilde{R}_h$, translation $\tilde{T}_h = [\tilde{t}_x, \tilde{t}_y]$ and camera parameter $\tilde{s}$.

### B.3 ACCURACY OF IRIS LABELS IN TRAINING

In this section, we will introduce the process of generating iris labels used in training. We train a separate model Iris-lmk-Net to automatically detect the iris landmarks, which are used as iris labels when training with constraints. Besides, we explored how iris label accuracy affect the training process and model performance and we tried to reduce the effect of possible inaccurate iris landmarks when the eye image is blurry or occluded.

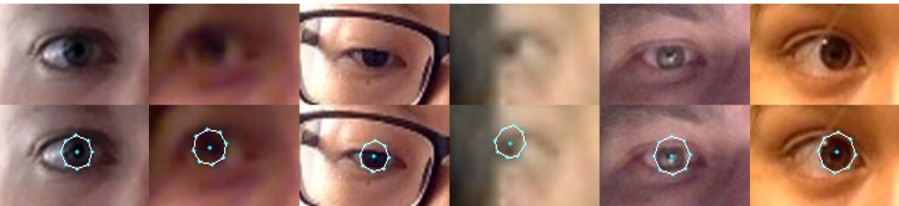

Figure 6: Iris landmark detection on testing images using Iris-lmk-Net.

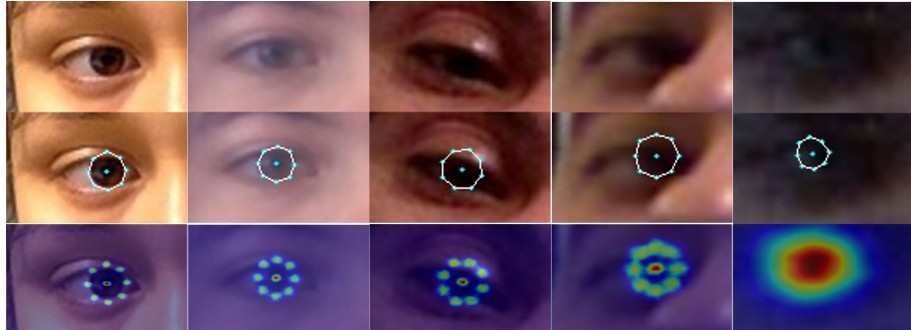

Figure 7: Iris landmark position and uncertainty map generated by Iris-lmk-Net. From left to right, iris landmark variance $\boldsymbol{\Sigma}_{\tilde{z}_i}$ is becoming larger due to illumination and iris visibility difference.

For training the gaze model, we propose to use iris re-projection loss for learning the eyeball shape and position. In order to obtain iris labels, we pre-train an Iris-lmk-Net for regressing iris landmarks from an eye image. We provide training details for Iris-lmk-Net below. The input eye image is passed through a ResNet18 architecture followed by a 3-layer MLP (multi-layer perceptron). Overall the iris landmarks contains eight boundary points and one iris center point. The Iris-lmk-Net estimates the mean $\{\boldsymbol{u}_{\tilde{z}_i} = [u_{x_i}, u_{y_i}]\}_{i=1}^{9}$ and Cholesky coefficients $\{\boldsymbol{L}_i\}_{i=1}^{9}$ of the covariance matrix for each iris landmark. Then the distribution for each landmark can be expressed by:

$$p(\tilde{\boldsymbol{z}}_i | \boldsymbol{u}_{\tilde{z}_i}, \boldsymbol{\Sigma}_{\tilde{z}_i}) = \mathcal{N}(\tilde{\boldsymbol{z}}_i | \boldsymbol{u}_{\tilde{z}_i}, \boldsymbol{\Sigma}_{\tilde{z}_i}), \text{where} \boldsymbol{\Sigma}_{\tilde{z}_i} = \boldsymbol{L}_{\tilde{z}_i} \boldsymbol{L}_{\tilde{z}_i}^T \tag{16}$$

We define a negative log-likelihood loss for training Iris-lmk-Net.

$$L_{nll} = -\frac{1}{9} \sum_{i}^{9} \log(\mathcal{N}(\boldsymbol{z}_i; \boldsymbol{u}_{\tilde{z}_i}, \boldsymbol{\Sigma}_{\tilde{z}_i}))$$

$$= \frac{1}{9 \times 2} \sum_{i}^{9} \left( \log |\boldsymbol{\Sigma}_{\tilde{z}_i}| + (\boldsymbol{z}_i - \boldsymbol{u}_{\tilde{z}_i})^T \boldsymbol{\Sigma}_{\tilde{z}_i}^{-1} (\boldsymbol{z}_i - \boldsymbol{u}_{\tilde{z}_i}) \right) \tag{17}$$

where $\boldsymbol{z}_i$ is the iris landmark detected by a public software Mediapipe. We train Iris-lmk-Net on a hybrid of gaze datasets and web face images, including Gaze360 (Kellnhofer et al., 2019), ETH-XGaze (Zhang et al., 2020) and CelebA (Liu et al., 2015). We train 50 epochs in total. In Fig. 6, we provide visualization results on testing dataset MPIIFace and EyeDiap.

Table 5: Training UGaze-Geo with iris landmark predicted by Mediapipe and our Iris-lmk-Net and results of cross-dataset evaluations.

| UGaze-Geo + iris | $\mathcal{D}_E \to \mathcal{D}_M$ | $\mathcal{D}_E \to \mathcal{D}_D$ | $\mathcal{D}_G \to \mathcal{D}_M$ | $\mathcal{D}_G \to \mathcal{D}_D$ |
|---|---|---|---|---|
| Iris from: Mediapipe | 7.04 | **9.80** | 7.38 | 6.95 |
| Iris from: Iris-lmk-Net | **6.92** | 9.84 | **7.23** | **6.87** |

We compare gaze estimation performance from UGaze-Geo by directly utilizing iris landmarks from Mediapipe and using our pre-trained Iris-lmk-Net respectively and summarize them in Table. 5. We also show that using the iris landmark distribution as supervision for training the model achieves better performance than directly applying iris landmarks from Mediapipe on tasks of $\mathcal{D}_E \to \mathcal{D}_M$, $\mathcal{D}_G \to \mathcal{D}_M$ and $\mathcal{D}_G \to \mathcal{D}_D$. Especially on Gaze360, wrong labels can be generated by Mediapipe due to image blur, low resolution or self-occlusion, then resulting in incorrect eyeball rotation learning. When training with the iris landmark distribution of $p(\tilde{z}_i | \boldsymbol{u}_{\tilde{z}_i}, \boldsymbol{\Sigma}_{\tilde{z}_i})$, the training samples with larger iris variances $\boldsymbol{\Sigma}_{\tilde{z}_i}$ will be assigned with less weight. In Fig. 7, we show the difference of predicted iris variances from Iris-lmk-Net based on illumination condition and iris visibility. For blurred images, or occluded eye region caused by large head pose, the predicted iris has larger uncertainty (reflected by the variance) and the mean position can be inaccurate (column 5 in Fig. 7).

## C  CROSS-DATASET PERFORMANCE ANALYSIS

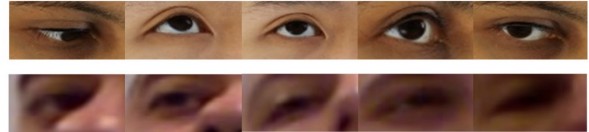

Figure 8: Image resolution shift between ETH-XGaze (row1) and EyeDiap (row2).

In Section 4.3 of our paper, we provide cross-dataset evaluation on four datasets. When testing On EyeDiap dataset ($\mathcal{D}_D$), our model achieves better accuracy than SOTA appearanc-based methods on the task of $\mathcal{D}_G \to \mathcal{D}_D$ while performs worse on the task of $\mathcal{D}_E \to \mathcal{D}_D$. The Performance drop in $\mathcal{D}_E \to \mathcal{D}_D$ can be caused by the data distribution shift between the training data and testing data: 1) image resolution shift: EyeDiap images has much lower resolution than ETH-XGaze and the eye region are more blurry, making it harder to recognize the iris feature during testing, as shown in Fig. 8; 2) head pose distribution shift: the training data ETH-XGaze have large head pose variations in pitch and yaw angles (not in roll angles), while EyeDiap images also vary in roll angles. It can be an essential factor since our model predicts relative eyeball rotation, which is a sequential result of head pose prediction. In addition, in the training data preparation process (for 3D head poses, camera factors and facial landmarks), the 3D reconstruction model or the facial landmark detector may fail to generate reliable results on images with extreme head poses or incomplete faces in ETH-XGaze, resulting in label noise in training. For this issue we will further refine the face alignment results on ETH-XGaze.

## D  COMPARE WITH OTHER DATA AUGMENTATION

As we create an augmented batch for each original training image to impose the geometric constraints. We perform a comparison of performance improvement over baseline model after using the augmented data. We compare with the 2D augmentation approaches in (Bao et al., 2022), as shown in Table. 6. (Bao et al., 2022) generated pseudo gaze labels during the "RAT" stage and considered another two image augmentations: Geometry and Noise. Numerically, data augmentation with 3D head rotations with our geometric constraints have better generalization ability on the tasks of $\mathcal{D}_E \to \mathcal{D}_M$, $\mathcal{D}_G \to \mathcal{D}_M$ and $\mathcal{D}_G \to \mathcal{D}_D$. We also show relative improvement in the last two rows

Table 6: comparing with 2D RAT (rotation augmented training) proposed in (Bao et al., 2022) and (Wang et al., 2022)

| Methods | $\mathcal{D}_E \to \mathcal{D}_M$ | $\mathcal{D}_E \to \mathcal{D}_D$ | $\mathcal{D}_G \to \mathcal{D}_M$ | $\mathcal{D}_G \to \mathcal{D}_D$ |
|---|---|---|---|---|
| Baseline(Bao et al., 2022) | 8.20 | **7.16** | 7.74 | 7.64 |
| Geometry+RAT(Bao et al., 2022) | 9.75 | 8.50 | 7.88 | 7.41 |
| Noise+RAT(Bao et al., 2022) | 8.70 | 8.12 | 7.80 | 7.65 |
| 2D Rotation+RAT(Bao et al., 2022) | 7.92 | 7.44 | 7.60 | 7.10 |
| Baseline (**ours**) | 8.27 | 10.77 | 8.90 | 9.66 |
| UGaze-Geo(**ours**) | **6.92** | 9.84 | **7.23** | **6.87** |
| Relative | 2D RAT : -3.41% | +3.91% | -1.81% | -7.07% |
| improvement | ours: **-16.30%** | **-8.35%** | **-18.76%** | **-40.61%** |

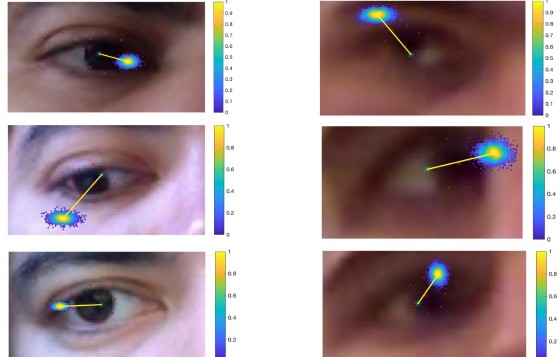

Figure 9: Gaze direction distribution, where the blue * and red * represents the projected eyeball center and iris center, the yellow line shows the average gaze direction. **Column1**: training on Gaze360 with testing samples from MPIIFace dataset, **column2**: training on Gaze360 with testing samples from EyeDiap dataset.

of Table 6, by computing the gaze error reduction percentage between rotation-augmented method and baseline. Although the performance of the baseline models in Table. 6 differs, our proposed data augmentation with applied constraints yields a larger measure of improvement over baseline model.

## E MORE ABOUT GAZE UNCERTAINTY

In this section we show more detailed gaze uncertainty analysis given the rotation distribution predicted by UGaze-Geo framework. Based on Section 3.3 in our paper, our model predicts a distribution $\mathcal{N}([\psi, \theta]^T; \boldsymbol{u}_e^{(n)}, \boldsymbol{\Sigma}_e^{(n)})$ for relative eyeball rotation angles. Then we can estimate the uncertainty of gaze by sampling from the distribution. As shown in Fig. 9, if we sample sufficient times and compute the corresponding gaze direction, we can generate an "ellipse like" density map indicating the distribution of possible gaze directions, where the major axis and minor axis representing the scale of gaze uncertainty in azimuth and elevation angle. When our model is trained on the same dataset, the gaze uncertainty may vary in direction, and vary with the visibility of iris and eye region.

