# OpenReview forum: "Weakly-supervised & Uncertainty-aware 3D Gaze Estimation with Geometry-guided Constraints"
_ICLR.cc/2025/Conference — Submitted to ICLR 2025_

### Official Review · Reviewer_pDez · 2024-10-26

**Soundness:** 3
**Presentation:** 3
**Contribution:** 2
**Rating:** 5
**Confidence:** 5

**Summary:**

This paper presents a novel framework for estimating 3D gaze that leverages weak supervision and integrates geometric constraints inspired by human eye anatomy. To tackle challenges such as changes in head pose, occlusions, and varying lighting conditions, the proposed framework separates gaze estimation into head and eye movements while adding uncertainty modeling for more robust predictions. The use of anatomical priors allows it to generalize well across different datasets. The authors show strong performance even when trained with limited labeled data.

**Strengths:**

Originality: This paper presents a new approach by combining weak supervision with geometric constraints inspired by human eye anatomy. This mix of anatomical knowledge with uncertainty-aware modeling is unique in gaze estimation. The focus on disentangling head and eye movements without extensive gaze labels is original and practical for real-world applications.

Quality: The technical quality is solid. The authors explain each part of the framework in detail and test it thoroughly across different datasets. The ablation study shows how each constraint improves the model.

Clarity: The methodology is explained clearly, and each loss function is well-defined. Visual examples of how the model works in difficult cases, like low light or occlusions, would help make it even clearer. Still, the explanations are mostly easy to follow.

Significance: This model could be important for applications in VR/AR, human-computer interaction, and driver monitoring, where gaze estimation is hard without accurate labels. Reducing label dependency and using anatomical constraints makes the model more flexible and useful across domains. The approach aligns well with trends in making machine learning more adaptable and data-efficient.

**Weaknesses:**

The paper combines weak supervision with geometric constraints, which is interesting. However, it could better clarify what is new compared to previous methods, especially other anatomy-based gaze models like Ploumpis et al. (2020). A more direct comparison with similar models would also help highlight UGaze-Geo’s unique features, like its uncertainty modeling. Explaining how this uncertainty approach differs from existing probabilistic methods could also strengthen the originality claims.

The reliance on pre-trained models for head pose and iris detection raises scalability issues. Errors from these models can impact final gaze estimates, but the paper does not address this risk. It would be useful to test how UGaze-Geo performs with noisy or imperfect pre-processing outputs. Additionally, assuming a consistent eye anatomy across subjects may limit generalization.

The cross-dataset testing is solid, but further analysis would add depth. For example, breaking down results by lighting or head pose conditions would show how well UGaze-Geo handles real-world challenges. Also, the head pose augmentation may not fully reflect realistic gaze patterns. Including a comparison with and without augmentation could clarify its practical impact.

UGaze-Geo has potential for real-world use, but there is no validation in real-time settings, where motion blur and occlusion are frequent. Additionally, the uncertainty metrics are promising, but it is unclear how they help manage prediction errors.

**Questions:**

- Given the extensive pre-processing (head pose, iris landmarks, etc.), please justify your claim that this is a weakly supervised approach?
- What is the actual impact of the errors from pre-trained models i(ead pose and iris detection) on gaze estimates?
- How robust are your geometric constraints (e.g., fixed eyeball center, constant radii) across individuals with varying anatomy? Can you provide some quantitative measures
- How do gaze ranges, environmental conditions, and resolution affect cross-dataset results? Have you analyzed these factors?
- Did you test the model for real-time adaptability in dynamic conditions like lighting changes or occlusions?
- How does reducing labeled data impact performance in both within- and cross-dataset tests?
- How well do your synthetic samples replicate natural gaze in varied, unconstrained settings?

---

### Official Review · Reviewer_oxqm · 2024-11-03

**Soundness:** 3
**Presentation:** 3
**Contribution:** 3
**Rating:** 6
**Confidence:** 5

**Summary:**

This paper proposes a so-called model-based gaze estimation approach that integrates 3D eyeball anatomy via careful definition of parameters and composition of predicted values. By incorporating eye landmark pseudo-labels (acquired via a model trained on Mediapipe predictions), 3D face/head augmentations using 3DDFA, and facial landmark pseudo-labels (acquired via FAN), the proposed work is able to train gaze estimators with zero or few gaze labels, thanks to explicitly modeling the eyeball. The authors show good performance in both within-dataset and cross-dataset experiments for gaze direction estimation. They also demonstrate the capability to model and predict uncertainties for iris landmark coordinates and gaze direction.

**Strengths:**

The overall approach is meticulously designed, and is refreshing given that many recent methods are designed to be naively-end-to-end (and cannot handle partially labeled data). The explicit modeling of the eyeball in relation to the head can be sensitive to the detected eye region landmarks, and the authors perform uncertainty-aware learning using high-quality Mediapipe predictions to address this issue (yielding their Iris-lmk-Net).

Performance-wise, the proposed method seems to out-perform state-of-the-art methods for several benchmarks in both the within and cross dataset settings.

Particularly impressive is how the Gaze-Geo method performs with as few as 25% of the gaze labels in the Gaze360 dataset (both cross-dataset and within-dataset performances are good).

**Weaknesses:**

The impact of the 3DDFA training augmentation is likely to be very huge. Similar work was shown in [1] where 3DDFA was also used to create novel head poses of the original data. In a way, this novel-view-synthesis could be considered as part of the proposed Gaze-Geo and UGaze-Geo methods. However, the 3DDFA-based 3D head pose augmentation at training time may be doing the heavy-lifting, and therefore the absence of an ablation study for this component is quite a shame. Specifically, it would be good to have an extra row in Table 4 which shows the results when using the dataset without any 3DDFA-based generation.

Secondly, the results shown in this paper do not seem to compare against [2] which out-performs the proposed method on several counts. As [2] was published at CVPR 2024, it was publicly available at the time of submission to ICLR 2025, and thus a comparison would be appropriate.

The authors do not discuss the limitations or societal impact of their work.

[1] Qin, Jiawei, Takuru Shimoyama, and Yusuke Sugano. "Learning-by-novel-view-synthesis for full-face appearance-based 3d gaze estimation." Proceedings of the IEEE/CVF Conference on Computer Vision and Pattern Recognition. 2022.

[2] Bao, Yiwei, and Feng Lu. "From Feature to Gaze: A Generalizable Replacement of Linear Layer for Gaze Estimation." Proceedings of the IEEE/CVF Conference on Computer Vision and Pattern Recognition. 2024.

**Questions:**

The presentation of the ablation study is quite confusing to be honest. Would it be possible to make it easier to understand? For example, instead of using “i-th constraint” it may help to refer to an acronym or the loss term.

---

### Official Review · Reviewer_aS5m · 2024-11-03

**Soundness:** 3
**Presentation:** 2
**Contribution:** 3
**Rating:** 5
**Confidence:** 5

**Summary:**

The paper introduces UGaze-Geo, a weakly-supervised framework for 3D gaze estimation that leverages geometric constraints related to eyeball anatomy, head rotation, and gaze. Experiments show that the model can achieve better within-dataset and cross-dataset performance on gaze estimation than previous end-to-end training methods.

**Strengths:**

1. The paper decomposes gaze direction into head rotation and eyeball rotation relative to the camera, leveraging physical principles for better estimation than end-to-end approaches.

2. The use of a re-projection loss for eye landmarks to supervise eyeball rotation is interesting.

**Weaknesses:**

1. The training details for the iris detector are unclear, including how iris labels are obtained and the hyperparameter settings used.

2. The experimental validation is insufficient:

    a. The paper lacks within-dataset results on ETH-XGaze, which would be valuable for exploring the disentanglement of head pose and eyeball rotation.

    b. The paper lacks the cross-dataset results on GazeCapture, which is collected in various in-the-wild environments.

    c. In Table 1, only one cross-dataset method (Bao et al., 2022) is used for comparison, and only its performance with 100% data is shown. Results with limited data would provide a more comprehensive comparison. Based on the results in Table 1, there is no clear advantage of the proposed method over Bao et al. (2022). Additional experiments could strengthen the paper's claims.

3. The accuracy of eyeball rotation and head pose estimation for the proposed method is not clearly presented. It remains unclear whether the method can effectively handle these two problems. The potential benefits of using a pre-trained head pose estimator are not explored or discussed.

4. The paper does not discuss why using uncertainty loss improves results, leaving a gap in understanding its impact on performance.

**Questions:**

The paper does not fully utilize the available 10-page limit, relegating important details to the supplementary material instead of including them in the main text.

---

### Official Review · Reviewer_oHLo · 2024-11-03

**Soundness:** 3
**Presentation:** 2
**Contribution:** 2
**Rating:** 5
**Confidence:** 5

**Summary:**

The paper presents a method for image-based 3D gaze estimation, in which gaze prediction is modeled as the composition of head pose and head-related eye rotation. The authors consider a spherical 3D geometry for eyes and based on that they design 3D anatomy aware losses which are used for weakly-supervised training. In addition, the authors employ a 3D rotation consistency loss between synthetic frames with different head pose but static head-eye rotation. The two auxiliary losses, combined with a probabilistic formulation, lead to lower errors compared to SOTA, in within- and cross-dataset experiments on common gaze estimation benchmark datasets.

**Strengths:**

1. The paper is overall well written and easy to follow and understand.
2. The uncertainty-aware model (UGaze-Geo) offers the chance to quantify prediction uncertainty which is often useful for applications. It is also shown by the results that outperforms the deterministic one (Gaze-Geo).
3. The effect of the proposed losses is demonstrated in an ablation study.
4. The method achieves good within- and cross-dataset performance, often outperforming the compared SOTA. Most importantly, the method can maintain accuracy using only a portion of the data.

**Weaknesses:**

1. The method relies on head pose prediction which might not be reliable when trained only in gaze datasets. Especially, as head-pose and texture variation is limited, cross-dataset performance could severely be affected because of that.
2. The method employs 3D eye geometry information in an implicit way (using few parameters) and designs losses around this idea. A recent method 3DGazeNet [1], predicts dense 3D eye structure with an end-to-end approach, demonstrating that dense 3D coordinate prediction is beneficial for gaze estimation and particularly in cross-dataset cases. This is a related work which should be discussed in the context of this work and included on the SOTA comparisons.
3. The quality of some visualizations could be improved. For example image alignment in a grid in Figure 2.
4. There is no evaluation of within-dataset results on ETH-XGaze and EyeDiap.

[1] Ververas, E., Gkagkos, P., Deng, J., Doukas, M.C., Guo, J. and Zafeiriou, S., 2025. 3DGazeNet: Generalizing 3D Gaze Estimation with Weak-Supervision from Synthetic Views. In European Conference on Computer Vision (pp. 387-404). Springer, Cham.

**Questions:**

1. Cross dataset experiments have been performed only between a dataset with wide head pose and gaze distribution and a dataset with narrow distributions. This is not totally the authors fault as this is what has mostly been done in relevant literature. For completeness, an important missing cross-dataset evaluation would be one between ETH-XGaze and Gaze360.

---

> ### Comment · Reviewer_oxqm · 2024-11-12
>
> I too thought of the 3DGazeNet [1] paper while reading this submission, but I would like to gently mention that 3DGazeNet is an ECCV 2024 paper which was published in late September - after the ICLR submission deadline. Therefore, the authors are not obligated to compare with or mention 3DGazeNet.

---

### Meta-Review · Area_Chair_yLzG · 2024-12-16

**Metareview:**

The paper introduces UGaze-Geo, a framework for weakly-supervised and uncertainty-aware 3D gaze estimation leveraging geometric constraints inspired by human eye anatomy. While the approach is innovative in combining anatomical knowledge with weak supervision, several significant weaknesses limit its overall contribution. Reviewers have highlighted concerns, including inadequate experimental validation, the absence of evaluations on key benchmarks like ETH-XGaze and EyeDiap, and missing implementation details such as iris detector hyperparameters and loss term definitions. Additionally, the authors have not provided responses to address these concerns, leaving them unresolved.

**Additional Comments On Reviewer Discussion:**

The authors did not provide response.

---

### Decision · Program_Chairs · 2025-01-22

Reject